# Facile Synthesis of Porous g-C_3_N_4_ with Enhanced Visible-Light Photoactivity

**DOI:** 10.3390/molecules27061754

**Published:** 2022-03-08

**Authors:** Guangyuan Yao, Yuqiang Liu, Jingcai Liu, Ya Xu

**Affiliations:** 1State Key Laboratory of Environmental Criteria and Risk Assessment, Chinese Research Academy of Environmental Sciences, Beijing 100012, China; slxw.yao1990@hotmail.com; 2Research Institute of Soil and Solid Waste, Chinese Research Academy of Environmental Sciences, Beijing 100012, China; liuyq@craes.org.cn

**Keywords:** porous g-C_3_N_4_, O doping, visible-light irradiation, photocatalysis

## Abstract

Porous graphitic carbon nitride (g-C_3_N_4_) was prepared by dicyandiamide and urea via the pyrolysis method, which possessed enhanced visible-light-driven photocatalytic performance. Its surface area was increased from 17.12 to 48.00 m^2^/g. The porous structure not only enhanced the light capture capacity, but also accelerated the mass transfer ability. The Di (Dicyandiamide)/Ur (Urea) composite possessed better photocatalytic activity for Rhodamine B in visible light than that of g-C_3_N_4_. Moreover, the Di/Ur-4:5 composite showed the best photoactivity, which was almost 5.8 times that of g-C_3_N_4_. The enhanced photocatalytic activity showed that holes and superoxide radical played a key role in the process of photodegradation, which was ascribed to the enhanced separation of photogenerated carriers. The efficient separation of photogenerated electron-hole pairs may be owing to the higher surface area, O dopant, and pore volumes, which can not only improve the trapping opportunities of charge carriers but also the retarded charge carrier recombination. Therefore, it is expected that the composite would be a promising candidate material for organic pollutant degradation.

## 1. Introduction

Nowadays, environmental pollution has become a worldwide problem of high concern, especially in developing countries. Therefore, many technologies have been applied to deal with this issue, such as chemical oxidation and reduction [1,2], adsorption [3,4], coagulation [5], extraction [6,7] and biological treatment [8]. However, these technologies have the disadvantages of secondary pollution and limited application. Therefore, photocatalysts have attracted more and more attention because of their utilization of abundant solar energy, having no need for additional regents, and no production of waste. The g-C_3_N_4_ was turned to be a novel organic semiconductor with excellent visible-light response because of its suitable band gap (2.7 eV), stable thermal and chemical properties, and low cost [9]. Generally, both low surface area and separation efficiency of photogenerated electron-hole pairs can depress the photoactivity of g-C_3_N_4_. Hence, various methods have been carried out to overcome these problems, such as coupling with semiconductors, metal oxide or polymers [10,11,12,13,14,15], ion doping [16,17,18,19], noble metal deposition [20,21,22,23], controlling morphology [24,25,26] and loading on carriers [9,27].

Among these methods, porous g-C_3_N_4_ has gained increasing attention because of its large surface area with more active sites, which could facilitate mass transfer ability and improve its photocatalytic performance. Generally, porous g-C_3_N_4_ can be obtained via soft-templating or silica-templating methods. As for the soft-templating methods, the carbon left in the products from the template might restrain its photoactivity. The silica-templating method is another important way to synthesize porous g-C_3_N_4_. However, it involves the further removal of silica by hydrogen fluoride, which is hazardous and toxic to the environment. Therefore, it is urgent to synthesize porous g-C_3_N_4_ facilely and friendly.

Generally, we use dicyandiamide or urea to synthesize g-C_3_N_4_. However, the very low yield of porous g-C_3_N_4_ from urea may limit its practical application. The bulk g-C_3_N_4_ with low surface area could be obtained from dicyandiamide, which greatly depresses its photocatalytic performance. Therefore, we obtain g-C_3_N_4_ by dicyandiamide, and use urea as a nontoxic bubble template, which can produce gas bubbles during the heating process and then form a porous structure in the target composites. The porous g-C_3_N_4_, synthesized via dicyandiamide and urea as co-precursors, is expected to have enhanced visible-light-driven photocatalytic performance [28].

In the current work, we gained the porous g-C_3_N_4_ by dicyandiamide and urea via the pyrolysis method. Then, we characterized the structure, morphology and physicochemical properties of the photocatalysts. Consequently, Rhodamine B was used as the target pollutant to assess its photocatalytic performance. Meanwhile, we studied the photocatalytic activity under visible light with a different mass ratio of dicyandiamide and urea. Furthermore, we also systematically investigated the improved photocatalytic performance, as well as the enhancement mechanism according to the above analysis.

## 2. Results and Discussion

### 2.1. XRD Analysis

The XRD patterns of Di/Ur composites and g-C_3_N_4_ were displayed in Figure 1. All samples exhibited the typical diffraction peaks at 2θ = 12.8° and 27.6°, which could be attributed to the interlayer stacking of aromatic systems and the in-plane structure motifs, respectively. Furthermore, the Di/Ur composites all possessed higher intensities than that of g-C_3_N_4_, and increased as the mass rates of urea and dicyandiamide increased.

### 2.2. SEM Analysis

Figure 2a,b display the morphology and structures of CN and the Di/Ur-4:5 composite, and it can be seen that the CN possessed bulk structure, which was formed by lamellar structures stacking with each other. However, the Di/Ur-4:5 composite possessed a loose structure and pores in its framework, which were beneficial for the photocatalytic performance.

Furthermore, the N_2_ adsorption–desorption isotherms of g-C_3_N_4_ and Di/Ur composites are shown in Figure 3. The adsorption–desorption isotherms of Di/Ur-4:5 composite exhibited a type IV curve with a small hysteresis loop, revealing the presence of mesoporous structure in the Di/Ur-4:5 composite. Then, the pore structure characterization of g-C_3_N_4_ and Di/Ur composites was also conducted, and the parameters were presented in Table 1. The as-synthesized Di/Ur samples possessed enhanced surface areas and pore volumes compared with those of CN, improving the adsorption capacity and providing more active sites for the photodegradation of Rh B [29,30,31].

### 2.3. XPS Analysis

The chemical bonding between the surface elements was determined. It demonstrated that the peaks of N 1s, C 1s and O 1s existed in both the Di/Ur-4:5 sample and g-C_3_N_4_. Moreover, the intensity of all spectra in the Di/Ur-4:5 sample was stronger than that of g-C_3_N_4_ (Figure 4a). The high-resolution spectra of O1s spectra (Figure 4b) can be divided into two peaks. The peaks at 531.6 and 532.8 eV should be attributed to the formation of the C-O-C species and the surface-adsorbed water, respectively [32]. This result confirmed that O atoms were bonded with C atoms in the basic structure of the photocatalyst. The C 1s spectra (Figure 4c) can be separated into four peaks. The peaks at 284.8, 286.3 and 288.2 eV should arise from the sp2 C-C bonds, C-NH_2_ species, and the sp2-hybridized carbon in the N-containing aromatic ring N-C-N, respectively [20]. Moreover, the weak peak at 289.5 eV also confirmed the creation of a C-O bond [33,34]. The N1s spectra (Figure 4d) exhibited peaks at 398.7, 400.1 and 404.5 eV, attributing to the pyridinic nitrogen of C=N-C, bridge N in N-[C]3N3 and π-excitations, respectively [33,35].

### 2.4. Photocatalytic Performance

The Rh B was used as the target pollutant to examine the photoactivities under visible-light irradiation (λ ≥ 420 nm). The results showed that all the Di/Ur composites exhibited better adsorption performance than g-C_3_N_4_ (Figure 5a), and it raised as the mass rate of urea increased. Generally, the adsorption sites on the surface of the sample increased with higher specific surface area and larger pore volumes, which enhanced its adsorption performance. These results corresponded to the BET analysis. Meanwhile, the photocatalytic performance of the Di/Ur composites were greatly enhanced, compared with g-C_3_N_4_, and increased as the mass rate of urea raised. Moreover, the Di/Ur-4:5 composite possessed the optimal photoactivities, and the photodegradation rate towards Rh B was almost 100% in 6 h under visible light. Therefore, we can infer that a porous structure with a greater surface area could improve mass transfer ability and photocatalytic performance, due to the more possible photocatalytic reaction sites exposed on its surface [9].

To get a deep understanding of the photocatalytic process, the kinetic experiments for Rh B photodegradation were conducted, and the results are displayed in Figure 5b. Generally, the pseudo-first order model can best describe the process of photodegradation [36]. The photoactivities of the obtained samples are as follows: Di/Ur-4:5 > Di/Ur-4:4 > Di/Ur-4:3 > Di/Ur-4:2 > g-C_3_N_4_. Meanwhile, it is apparent that all the Di/Ur composites possessed higher constants k than that of g-C_3_N_4_. The Di/Ur-4:5 sample exhibited the highest reaction rate constant, which was 5.8 times as much as that of g-C_3_N_4_. The high reaction rate constant may be attributed to the rising surface area, as depicted in Table 1.

We also investigated the stability and reusability of the Di/Ur-4:5 sample (Figure 6). It can be seen that the Di/Ur-4:5 sample showed excellent reusability during the photocatalytic reaction, and there was no significant deactivation even after four reaction runs. The slight descendant in the fourth run was attributed to intermediate poison on the surface of photocatalysts, which will lower the electron transfer velocity [37].

### 2.5. Optical Properties

Generally, the optical adsorption properties of the photocatalyst have a great effect on the photocatalytic performance. The UV-vis optical absorption spectra of the Di/Ur composites and g-C_3_N_4_ are displayed in Figure 7.

The results showed that the whole samples displayed excellent optical adsorption from UV light to visible light. The adsorption edge was at around 450 nm, and then extended to the region near 600 nm, which can be assigned to its smooth cleavage plane. This indicated that the porous structure may improve the utilization efficiency of visible light and then enhance its photoactivities (Figure 7a). Furthermore, the band gaps of the Di/Ur composites and g-C_3_N_4_ are presented in Figure 7b. The band gaps of Di/Ur composites (2.72 eV) were a little larger than that of g-C_3_N_4_ (2.67 eV).

To further make clear the behaviors of the photogenerated electron-hole carriers in the photocatalyst, the photoluminescence spectra were recorded. Herein, we present the PL spectrum of the Di/Ur-4:5 composite and g-C_3_N_4_. In general, PL intensity determined the recombination efficiency of the photogenerated charge carriers. The lower the PL intensity, the lower the recombination rate. The PL intensity of the Di/Ur-4:5 composite extremely decreased compared with that of g-C_3_N_4_ (Figure 8). Moreover, the separation efficiency of the photogenerated electron-hole carriers in the Di/Ur-4:5 composite was significantly accelerated under visible-light irradiation, which was probably attributed to its porous structure and the O dopant. This was consistent with the previous literatures [9,27].

Moreover, the transient photocurrent responses and Nyquist impedance plots (EIS) of g-C_3_N_4_ and Di/Ur-4:5 composite were also investigated to find out the photogenerated charge separation and electron transfer performance (Figure 9a,b). Notably, the photocurrent sharply declined when the light turned off, and then returned to fixed values when the light turned on. It implied that most photogenerated electrons transferred to the ITO substrates to generate photocurrent under visible-light irradiation. Furthermore, the Di/Ur-4:5 composite showed higher photocurrent intensity than that of g-C_3_N_4_, suggesting the higher separation rate of photogenerated charge carriers in the Di/Ur-4:5 composite. Additionally, the photocurrent can reproducibly increase and recover in every on–off cycle of irradiation, demonstrating the high stability in practical applications. Meanwhile, the arc radius of the Di/Ur-4:5 composite was smaller than that of g-C_3_N_4_ on the EIS Nyquist plot, which implied that the porous structure changed its charge distribution and made charge transfer easier [28]. The efficient separation of the photogenerated electron-hole pairs may be ascribed to the increased surface area and the enhanced redox potentials, which improved the trapping opportunities of the charge carriers by Rh B and retarded the charge carrier recombination due to the reduced spatial overlap [38]. Meanwhile, the O dopant may modulate the electronic structure and greatly enhance its separation rate [39]. These results are also consistent with our PL analysis.

### 2.6. Possible Mechanism

Generally, active species, such as •O_2_^−^, holes and·OH radicals, are generated by visible-light irradiation and suspected to be involved in the photocatalytic degradation reaction. To clarify the possible mechanism of the enhanced photoactivity of the Di/Ur-4:5 composite, we used BQ, TBA and EDTA-2Na as scavengers to identify their roles during its photocatalytic process. The photocatalytic degradation rate of Rh B was almost 100% without the addition of the scavengers (Figure 10a). When we added BQ and EDTA-2Na, the photocatalytic performance of Di/Ur-4:5 over Rh B notably declined, which indicated that •O_2_^−^ and holes play a significant role in the photoactivities. However, when we added TBA, the photocatalytic performance of Rh B slightly depressed, which revealed that •OH were a minor active species. It was concluded that both •O_2_^−^ and holes were major active species in the photodegradation of Rh B.

We also used the ESR technique to determine the active species during the photocatalytic process. It can be seen that there were no ESR signals of •O_2_^−^ and •OH species with DMPO in the dark (Figure 10b). Then, a significant evolution of ESR signal in DMSO was observed under visible light, demonstrating the important role of •O_2_^−^. However, no obvious ESR signal in H_2_O were observed, indicating the minor contribution of •OH. There results also corresponded to the radical trapping experiments. Therefore, combined with the above analysis, it was confirmed that both holes and •O_2_^−^ play a significant role in the photocatalytic process.

Based on the above analysis, the potential electron transfer route and possible mechanism of Di/Ur composites (Figure 11) on the photodegradation of Rh B was proposed. First, the obtained Di/Ur composite can easily absorb visible light because of the porous structure. Second, electrons (e^−^) will be transferred from the valence band (VB) to the conduction band (CB), and then create holes (h^+^) in the VB. Meanwhile, O dopant, with a higher surface area and larger pore volumes made the photogenerated electron-hole pairs more efficient to migrate, which could retard charge carrier recombination rate due to the reduced spatial overlap, and improve the trapping opportunities of charge carriers by reactants. Then, the adsorbed O_2_ could react with e^−^ to produce enough •O_2_^−^. Furthermore, the active sites needed for the adsorption and photocatalytic reaction can also be offered by the porous structure. In conclusion, the main active radicals •O_2_^−^ and the holes generated in the photocatalyst can effectively decompose the target pollutant (RhB) into ultimate products (CO_2_ and H_2_O) and other intermediates under visible-light irradiation.

## 3. Materials and Methods

### 3.1. Materials

Dicyandiamide (C_2_H_4_N_4_, AR, 98.5%/wt%), rhodamine B (C_28_H_31_ClN_2_O_3_, AR, 98.5%/wt%), urea (CH_4_N_2_O, AR, 99.0%/wt%), edetate disodium (EDTA-2Na, AR, 99.0%/wt%), benzoquinone (BQ, AR, 98.0%/vol%) and tert-butanol (TBA, GR, 99.5%/wt%). Distilled water was used in all the experiments.

### 3.2. Preparation of Di/Ur Composites

Firstly, we finely grounded the mixture of dicyandiamide and urea via mortar. Secondly, the obtained mixture was annealed for 4 h at 550 °C. After cooling down to the room temperature, the yellow powder was finely grounded and then heated at 500 °C for 2 h. Then, we collected the final resulting yellow products for further characterization and photocatalytic measurements. Finally, according to the different mass rates of dicyandiamide and urea, we defined the obtained Di/Ur composites as Di/Ur-4:2, Di/Ur-4:3, Di/Ur-4:4 and Di/Ur-4:5, respectively.

Similarly, we gained the g-C_3_N_4_ (CN) by dicyandiamide via the pyrolysis method at the same thermal conditions.

### 3.3. Characterization

X-ray diffraction (XRD) was recorded on a D8 advance X-ray diffractometer (Bruker, Billerica, MA, USA) equipped with Cu-Kα radiation (λ = 0.154056 nm) to identify the crystalline phase of the obtained photocatalysts. The samples were scanned in the range of 2θ from 10° to 80° with 0.02° step, at a scanning speed of 4°/min. An S-4800 scanning electron microscopy (Hitachi, Japan) was applied to investigate the surface morphology of the samples. The surface area of samples was performed by N_2_ adsorption at 77 K on a constant volume adsorption apparatus (JW-BK, JWGB Sci. and Tech., Beijing, China) and calculated by the Brunaer–Emmett–Teller (BET) method. The photoluminescence spectra were measured on a Hitachi F-7000 fluorescence spectrophotometer with an excitation wavelength of 400 nm for all the samples. The optical properties of the as-prepared samples were investigated by UV-vis diffuse reflectance spectroscopy (DRS) using a UV-vis spectrophotometer (U-3010, Hitachi, Toyko, Japan), where BaSO_4_ was used as the reference. The band gap values were calculated by extrapolating the linear part of the plot of (F(R)hν)^1/2^ versus hν: F(R)hν = A(hν − Eg)^2^, where F(R) = (1 − R)^2^/2R stands for the Kubelka–Munk function calculated from the reflectance spectrum, and hν is the photon energy expressed in eV. The electron spin resonance (ESR) signals of radical spin trapped by DMPO was at an ambient temperature on a JEOL (FA-200) spectrometer under visible-light irradiation of the suspension (0.05 mg/mL photocatalyst, 100 mM DMPO). The settings for ESR measurements were as follows: center field 324.019 mT, microwave frequency 9053.727 MHz, and power 0.99800 mW. Finally, the photocurrent and electrochemical impedance spectroscopy (EIS) of g-C_3_N_4_ and Di/Ur-4:5 composite were measured in a 0.1 M Na_2_SO_4_ aqueous solution with an electrochemical analyzer (CHI-660B, Shanghai, China). X-ray photoelectron spectroscopy (XPS) analysis was performed on the photoelectron spectrometer (Thermo Scientific Escalab 250Xi, Waltham, MA, USA) using monochromatic Al Kα radiation energy (λ = 1486.6 eV). Binding energies for the high-resolution spectra were calibrated by setting C 1 s at 284.8 eV.

### 3.4. Photoactivity Measurements

We used Rh B as a target pollutant to assess the photocatalytic performance of the as-synthesized samples under a 600 W Xenon lamp with a cut-off filter of 420 nm. First of all, 0.2 g of the photocatalysts was put into 100 mL of Rh B (10 ppm) aqueous solution and then magnetically stirred in the dark for 1 h to reach the adsorption–desorption equilibrium. Sequentially, 3 mL of suspension was collected at a certain time interval and centrifuged to remove photocatalyst particles for further analysis. Finally, the photodegradation effect was identified via a UV-vis spectrophotometer. We also used g-C_3_N_4_, Di/Ur-4:2, Di/Ur-4:3, Di/Ur-4:4 and Di/Ur-4:5 as references, and conducted comparative experiments under the same conditions. In addition, all the experiments were carried out in triplicates.

## 4. Conclusions

In summary, the porous g-C_3_N_4_ was successfully obtained via the pyrolysis method. The SEM, BET and XPS analyses revealed that the prepared samples possessed porous structures and O dopants, which not only enhanced light capture capacity but also accelerated mass transfer ability. The visible-light photoactivities of the obtained Di/Ur composites were greatly enhanced. Moreover, the Di/Ur-4:5 composite possessed the optimal photocatalytic performance towards RhB, which was almost 5.8 times that of g-C_3_N_4_. Meanwhile, it also exhibited exceptional stability and reusability. It was indicated that •O_2_^−^ and holes were the major active species, and the excellent photocatalytic performance in visible light was attributed to the enhanced separation of photogenerated carriers. The efficient separation rates may be ascribed to the O dopant and the improved pore structure, which improved the trapping opportunities of charge carriers by RhB and retarded the charge carrier recombination. Therefore, this study not only shed light on the facile construction of porous g-C_3_N_4_, but also showed great potential in the fields of the organic pollutant degradation.

## Figures and Tables

**Figure 1 molecules-27-01754-f001:**
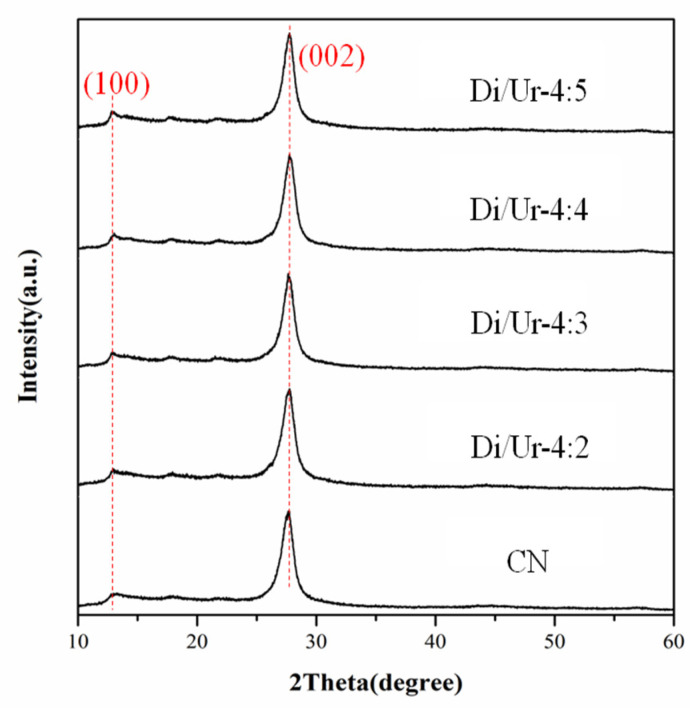
XRD patterns of g-C_3_N_4_ and Di/Ur composites.

**Figure 2 molecules-27-01754-f002:**
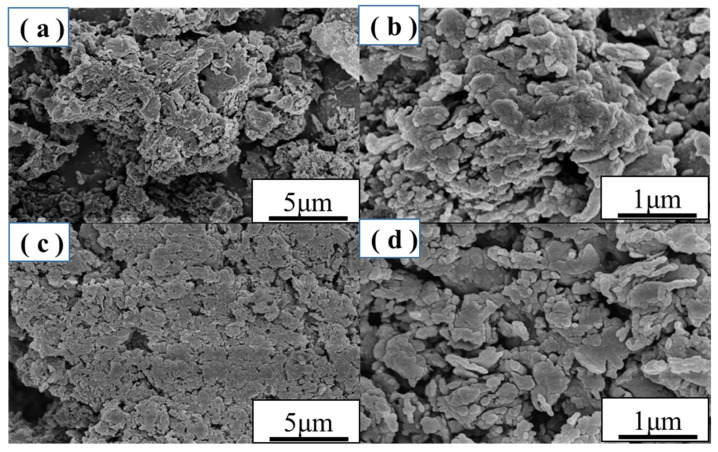
SEM images of (**a**,**b**) g-C_3_N_4_ and (**c**,**d**) Di/Ur-4:5 composite.

**Figure 3 molecules-27-01754-f003:**
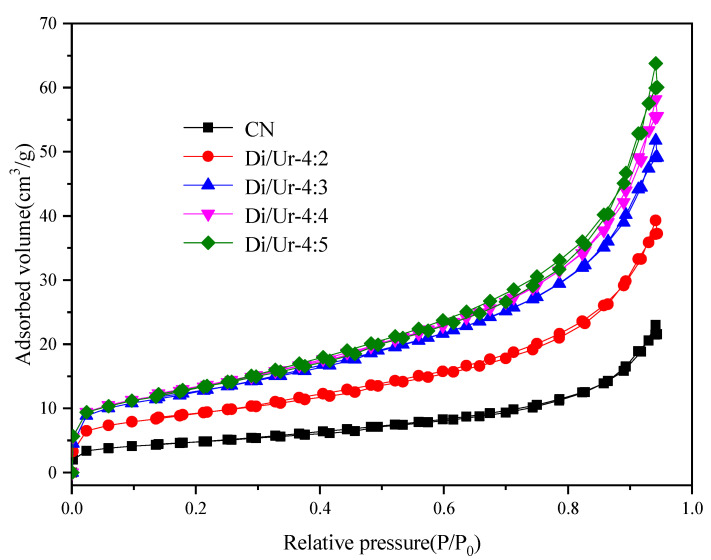
Nitrogen adsorption–desorption isotherms of g-C_3_N_4_ and the as-prepared Di/Ur composite.

**Figure 4 molecules-27-01754-f004:**
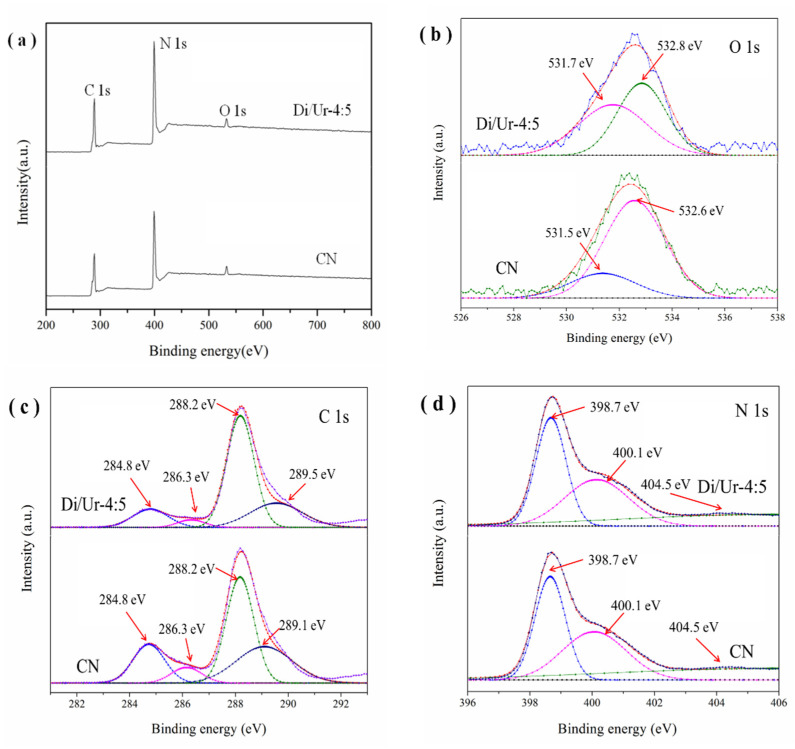
The XPS of Di/Ur-4:5 and CN for (**a**) survey spectrum; (**b**) O 1s; (**c**) C 1s; (**d**) N 1s.

**Figure 5 molecules-27-01754-f005:**
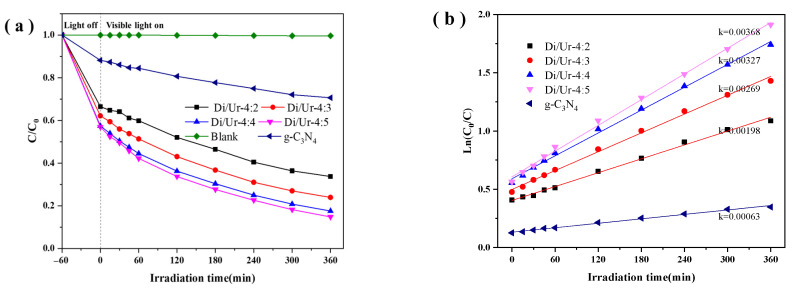
Photocatalytic degradation (**a**), and linear transform Ln(C_0_/C) of the kinetic curves (**b**), of RhB under visible light.

**Figure 6 molecules-27-01754-f006:**
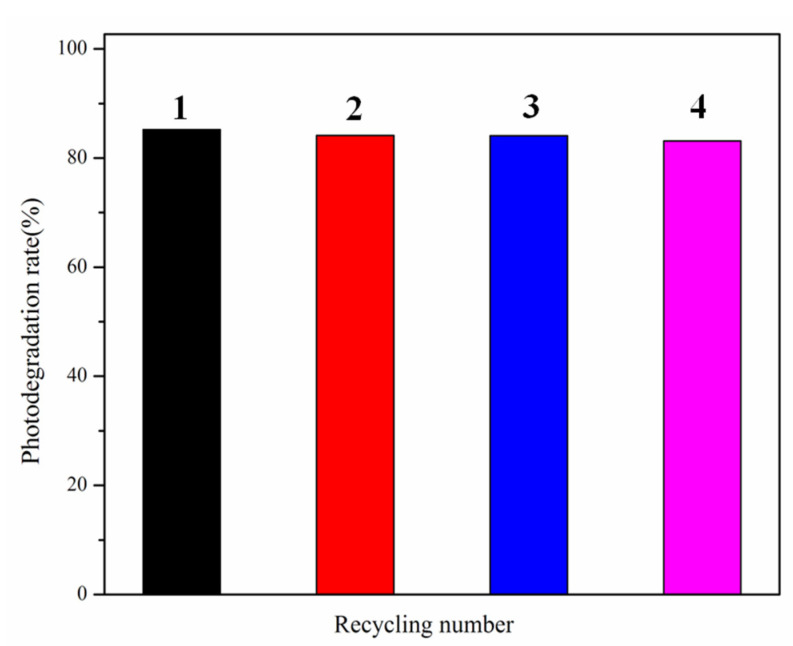
Recycling tests for the photodegradation of RhB over the Di/Ur-4:5 composite under visible light.

**Figure 7 molecules-27-01754-f007:**
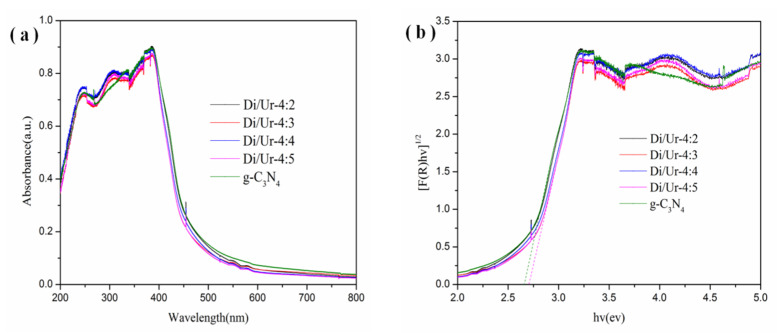
UV-Vis DRS (**a**), and band gaps (**b**), of g-C_3_N_4_ and the as-prepared Di/Ur composites.

**Figure 8 molecules-27-01754-f008:**
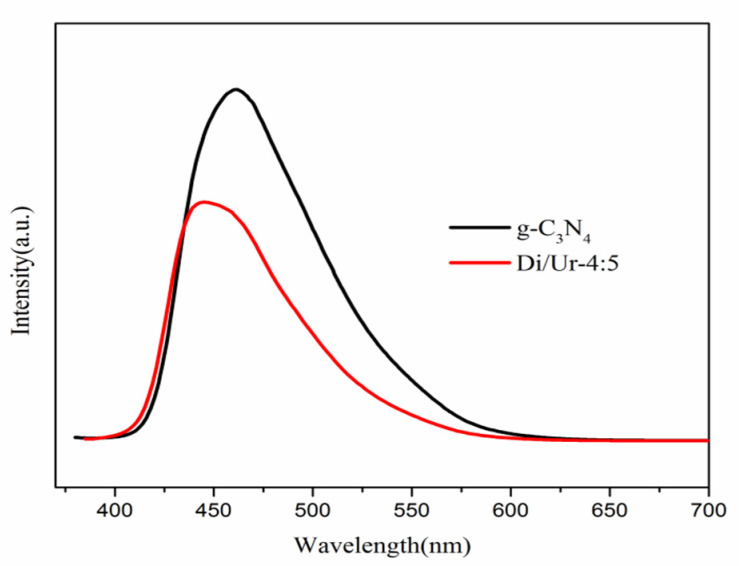
Photoluminescence (PL) spectra of pure g-C_3_N_4_ and the Di/Ur-4:5 composite.

**Figure 9 molecules-27-01754-f009:**
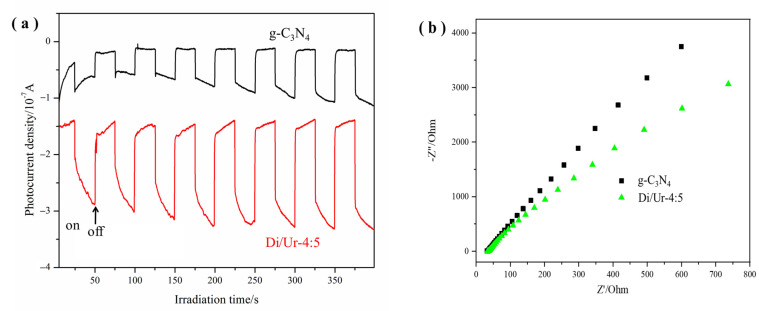
Transient photocurrent responses (**a**), and EIS Nyquist plots (**b**), of pure g-C_3_N_4_ and Di/Ur-4:5 composite.

**Figure 10 molecules-27-01754-f010:**
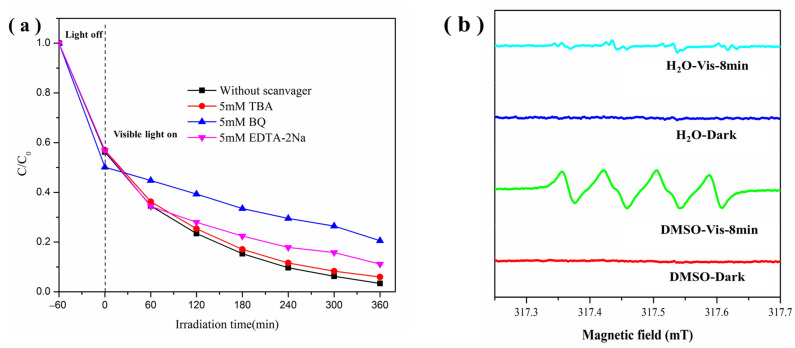
The plots of photogenerated carriers trapping on the photodegradation of RhB under visible light in the presence of the Di/Ur-4:5 composite (**a**), and ESR spectra of the Di/Ur-4:5 composite in DMSO solvents and water (**b**), with DMPO.

**Figure 11 molecules-27-01754-f011:**
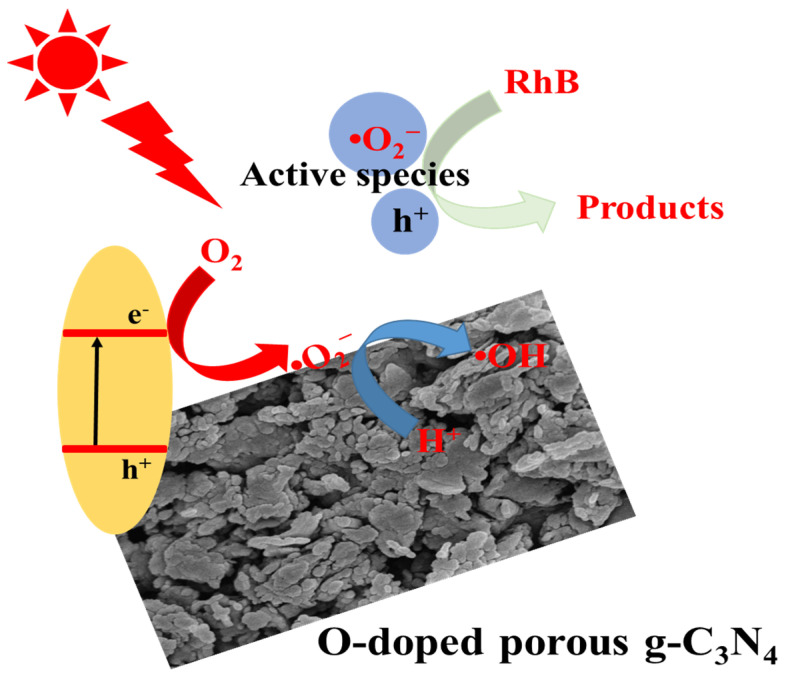
Schematic illustration of the enhanced photocatalytic mechanism of Di/Ur composites under visible-light irradiation.

**Table 1 molecules-27-01754-t001:** BET parameters of g-C_3_N_4_ and the obtained Di/Ur samples.

Samples	BET Surface Area (m^2^/g)	Pore Volume (cm^3^/g)	Average Pore Diameter (nm)
CN	17.115	0.038	8.315
Di/Ur-4:2	32.483	0.065	7.490
Di/Ur-4:3	44.694	0.085	7.167
Di/Ur-4:4	46.919	0.096	7.675
Di/Ur-4:5	48.002	0.107	8.220

## Data Availability

Not applicable.

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
