# Peer review of "Facile Synthesis of Porous g-C3N4 with Enhanced Visible-Light Photoactivity"

_molecules, 2022, doi:10.3390/molecules27061754_

Round 1

Reviewer 1 Report

In this article, the authors explain how the different mass rates of dicyandiamide and urea, used to synthesise porous graphitic carbon nitride, affect its physicochemical properties and catalytic activity. The physicochemical characterisation based on XRD, SEM, BET and XPS analysis. Photocatalytic activity of g-C3N4 was determined by Rhodamine B (RhB) oxidation, including recycling test. Comprehensive studies (ESR technique, EIS measurements, PL measurements) allowed authors to propose RhB photodegradation mechanism on g-C3N4. This study can help to expand the the-state-of-art in the catalytic field. I recommend this article to publish after some consideration: check file.

Author Response

Reviewer1# In this article, the authors explain how the different mass rates of dicyandiamide and urea, used to synthesise porous graphitic carbon nitride, affect its physicochemical properties and catalytic activity. The physicochemical characterisation based on XRD, SEM, BET and XPS analysis. Photocatalytic activity of g-C3N4 was determined by Rhodamine B (RhB) oxidation, including recycling test. Comprehensive studies (ESR technique, EIS measurements, PL measurements) allowed authors to propose RhB photodegradation mechanism on g-C3N4. This study can help to expand the the-state-of-art in the catalytic field. I recommend this article to publish after some consideration: 1. In figure 4a and 4b, colours corresponding to composites should be the same. Different colours might confuse readers. Answer: Thanks for the reviewer’s good advice. We have changed the colours of figure 4a and 4b to the same. Please check the revised version. 2. Based on the same graphs, seems like 40% of your RhB is rather adsorbed on the surface. I wonder if after reaction is still some RhB adsorbed on the surface?? If is the case, we cannot talk about 90% degradation. Answer: Thanks for the reviewer’s good advice. The suspension was collected and centrifuged at the end of the reaction, we find the colour of the photocatalyst is totally not red. Therefore, we talk about 90% degradation. 3. I would like author to comment what caused the blue-shifting in UV-Vis spectra (marked with narrow)? Is it suggesting that the C–N bonds in the nanocomposites Di/Ur are stronger than those in the parent material and influence d-d transition? Answer: Thanks for the reviewer’s good advice. As we know, the change of molecular conjugation degree will cause the shift of spectrum. Generally, the ultraviolet absorption spectrum of molecules will have a greater degree of blue shift with lower conjugation. Because there are many reasons that affect the degree of conjugation, we can just conclude that the C–N bonds in the nanocomposites Di/Ur are stronger than those in the parent material and then influence the optical absorption performance. 4. Caption to Figure 9b. ‘’is written and ESR spectra of the Di/Ur-4:5 composite in ….. and water’’, should provide information that is water with DMPO? As is written in text. Answer: Thanks for the reviewer’s good advice. We have provided the information that is water with DMPO. Please check the revised version.

Reviewer 2 Report

The paper describes a pyrolysis synthesis method to obtain graphitic carbon nitride photocatalysts for the degradation of organic pollutants.  The following comments should be addressed in the revised version:

1) Line 13. All abbreviations in the abstract should be avoided or explained.

2) Line 54 “. And the porous g-C3N4 synthesized via dicyandiamide and urea as co-precursors are expected to be with enhanced visible light-driven photocatalytic performance”. This should be explained or supported by a reference.

3) Line 58. “And then we characterized the structure, morphology and physicochemical properties of the photocatalysts were by XRD, ESR, SEM, BET, EIS, PL and UV-Vis spectra, respectively.” This needs to be mentioned in the abstract and in the experimental section rather than in the introduction.

Every equipment and protocol should be described. Mention XRD scanning rate and type of irradiation, mention parameters of the SEM microscope, type of equipment used in N2 chemisorption, and pretreatment procedure and other methods. XPS equipment and procedures are not mentioned at all in the experimental section.

4) Line 67. The purity of all chemicals should be reported.

5) Line 81. Specify the dimensions and the material of the reaction vessel.

6) Line 86. Specify the resolution and time to record the spectrum in the UV-vis spectrophotometer. Also specify its manufacturer.

7) Line 112. The accuracy of surface area value with two decimals does not look realistic. Please report an experimental error for these measurements. Add adsorption and desorption isotherms to a separate Figure.

8) Line 124.  The peaks at 284.8, 286.3 and 288.2 eV should arise from sp2 C-C… Please provide a supporting reference.

9) Line 133. “and it raised as the mass rate of urea increased”. Did you mean “mass ratio” ?

10) Table 2 can be removed. Fitted equation should be removed and instead the kinetic constant can be added as an additional column in Table 1.

11) Figures 4 and 5. Add experimental conditions (volume of solution, light intensity in mW/cm2 etc.) to the Figure captions.

12) Lines 145-152. Discussion should explain the order of kinetic constants based on the phiciso-chemical properties of the samples. The discussion should describe the “structure- activity” relationship.  Therefore it is better for this discussion to combine Table 1&2 as outlined in the previous comment.

13) The abbreviation in the y-axis title is not explained in Figure 6b. Also in the text, Figures 6a and Figure 6b should be described separately.

14) Figure 6. Could you replace a.u. with real units of absorbtion or add an absorption scale bar?

15) Figure 10 needs to be made more clear. Remove the blue background which obstructs the view.

Author Response

Reviewer2#

The paper describes a pyrolysis synthesis method to obtain graphitic carbon nitride photocatalysts for the degradation of organic pollutants. The following comments should be addressed in the revised version:

1) Line 13. All abbreviations in the abstract should be avoided or explained.

Answer: Thanks for the reviewer’s good advice. We have replaced or explained all the abbreviations in the abstract. Please check the revised version.

2) Line 54 “. And the porous g-C3N4 synthesized via dicyandiamide and urea as co-precursors are expected to be with enhanced visible light-driven photocatalytic performance”. This should be explained or supported by a reference.

Answer: Thanks for the reviewer’s good advice. We have added references in the sentence of Line 54 “And the porous g-C3N4 synthesized via dicyandiamide and urea as co-precursors are expected to be with enhanced visible light-driven photocatalytic performance”. Please check the revised version.

3) Line 58. “And then we characterized the structure, morphology and physicochemical properties of the photocatalysts were by XRD, ESR, SEM, BET, EIS, PL and UV-Vis spectra, respectively.” This needs to be mentioned in the abstract and in the experimental section rather than in the introduction. Every equipment and protocol should be described. Mention XRD scanning rate and type of irradiation, mention parameters of the SEM microscope, type of equipment used in N2 chemisorption, and pretreatment procedure and other methods. XPS equipment and procedures are not mentioned at all in the experimental section.

Answer: Thanks for the reviewer’s good advice. We have added every equipment and protocol in the experimental section. Please check the revised version.

4) Line 67. The purity of all chemicals should be reported.

Answer: Thanks for the reviewer’s good advice. We have reported the purity of all chemicals. Please check the revised version.

5) Line 81. Specify the dimensions and the material of the reaction vessel.

Answer: Thanks for the reviewer’s good advice. We did not specify the dimensions and the material of the reaction vessel. But we added experimental conditions (volume of solution etc.) in section of 2.4 Photoactivity measurements. Please check the revised version.

6) Line 86. Specify the resolution and time to record the spectrum in the UV-vis spectrophotometer. Also specify its manufacturer.

Answer: Thanks for the reviewer’s good advice. We have specified the manufacturer, resolution and time to record the spectrum in the UV-vis spectrophotometer. Please check the revised version.

7) Line 112. The accuracy of surface area value with two decimals does not look realistic. Please report an experimental error for these measurements. Add adsorption and desorption isotherms to a separate Figure.

Answer: Thanks for the reviewer’s good advice. We have revised the BET parameters, gave the information of the apparatus, and added adsorption and desorption isotherms to a separate Figure. Please check the revised version.

8) Line 124.  The peaks at 284.8, 286.3 and 288.2 eV should arise from sp2 C-C… Please provide a supporting reference.

Answer: Thanks for the reviewer’s good advice. We have provided a reference supporting “The peaks at 284.8, 286.3 and 288.2 eV should arise from sp2 C-C…”. Please check the revised version.

9) Line 133. “and it raised as the mass rate of urea increased”. Did you mean “mass ratio” ?

Answer: Thanks for the reviewer’s good advice. “and it raised as the mass rate of urea increased”, here it means “mass ratio”. Please check the revised version.

10) Table 2 can be removed. Fitted equation should be removed and instead the kinetic constant can be added as an additional column in Table 1.

Answer: Thanks for the reviewer’s good advice. We have removed the Table 2, and added the kinetic constant as an additional column in Figure 1. Please check the revised version.

11) Figures 4 and 5. Add experimental conditions (volume of solution, light intensity in mW/cm2 etc.) to the Figure captions.

Answer: Thanks for the reviewer’s good advice. We have added experimental conditions (volume of solution etc.) in section of 2.4 Photoactivity measurements. Please check the revised version.

12) Lines 145-152. Discussion should explain the order of kinetic constants based on the phiciso-chemical properties of the samples. The discussion should describe the “structure- activity” relationship. Therefore it is better for this discussion to combine Table 1&2 as outlined in the previous comment.

Answer: Thanks for the reviewer’s good advice. We have explained the order of kinetic constants based on the phiciso-chemical properties of the samples and combined the discussion with Table 1&2. Please check the revised version.

13) The abbreviation in the y-axis title is not explained in Figure 6b. Also in the text, Figures 6a and Figure 6b should be described separately.

Answer: Thanks for the reviewer’s good advice. We have explained the abbreviation of y-axis title in the chapter of 2.3. Characterization, and described them separately. Please check the revised version.

14) Figure 6. Could you replace a.u. with real units of absorbtion or add an absorption scale bar?

Answer: Thanks for the reviewer’s good advice. The optical properties of the as-prepared samples were investigated by UV-vis diffuse reflectance spectroscopy (DRS) using a UV-vis spectrophotometer (U-3010, Hitachi), where BaSO4 was used as the reference. Therefore,We used unit absorbance a.u.. Please check the revised version.

15) Figure 10 needs to be made more clear. Remove the blue background which obstructs the view.

Answer: Thanks for the reviewer’s good advice. We have removed the blue background and made Figure 10 more clear. Please check the revised version.

Reviewer 3 Report

The work is interesting however requires major revision.

Introduction requires careful revision with the recent references based on C3N4.

The importance of the work can be shown by taking some state of the art developments.

The table should be provided for comparison and recycled tests with stability characterization like XRD or SEM.

EIS measurement in dark and light has a light requires further evaluation for final material Di/Ur-4:5 composite and provides the justification with the suitable references.

The band-gap should be summarized using the plots in the table.

Author Response

Reviewer3#

The work is interesting however requires major revision.

Introduction requires careful revision with the recent references based on C3N4.

Answer: Thanks for the reviewer’s good advice. We have revised the introduction carefully and added recent references based on C3N4. Please check the revised version.

The importance of the work can be shown by taking some state of the art developments.

Answer: Thanks for the reviewer’s good advice. We have revised the introduction carefully according your suggestion. Please check the revised version.

The table should be provided for comparison and recycled tests with stability characterization like XRD or SEM.

Answer: We did the recycling tests to investigate the stability and reusability of the Di/Ur-4:5 sample, and the recycling tests indicated that there is no significant deactivation even after four reaction runs. Furthermore, we reviewed many literatures, and they also just did the recycling tests. Therefore, we did not do other characterization like XRD or SEM. Please check the revised version.

EIS measurement in dark and light has a light requires further evaluation for final material Di/Ur-4:5 composite and provides the justification with the suitable references.

The band-gap should be summarized using the plots in the table.

Answer: Thanks for the reviewer’s good advice. From the EIS data, the Di/Ur-4:5 composite possessed higher separation rate of photogenerated charge carriers in dark and light situation. We have changed relative figure. Meanwhile, we also summarized the band-gap. Please check the revised version.

Round 2

Reviewer 2 Report

Most of my comments have been addressed. 

The remaining comments are:

Specify percentage whether this is vol% or wt% when describe the purity of all chemicals.

The information on the light intensity is not provided. The volume of the liquid is mentioned, however the illuminated surface area is missing. 

English needs to be checked. There are many typos and grammatical errors. Just a few examples:

line 65. Rhodamine B were used as the target pollutant, should be "was used"

line 143. "a type â…£ cure"   should be “curve”

line 144.  "the pore structure characteristic ...were conducted" should be "characterization ... was conducted" 

line 176 “ the adsorption sites on the surface of the sample will increase with higher specific surface area and larger pore volume, which enhanced…”

Future and past tense should be avoided in the same sentence.

Line 182  “we can inferred”

Author Response

Reviewer2#

Most of my comments have been addressed. The remaining comments are:

  1. Specify percentage whether this is vol% or wt% when describe the purity of all chemicals.

Answer: Thanks for the reviewer’s good advice. We specified the percentage when describe the purity of all chemicals. Please check the revised version.

  1. The information on the light intensity is not provided. The volume of the liquid is mentioned, however the illuminated surface area is missing.

Answer: Thanks for the reviewer’s good advice. We have provided the light intensity in chapter 2.4. Photoactivity measurements (We used Rh B as target pollutant to assess the photocatalytic performance of as-synthesized samples under a 600 W Xenon lamp with a cut-off filter of 420 nm.). We did not add the illuminated surface area, because the expression of light in our paper is consistent with the contents of other papers. Please check the revised version.

  1. English needs to be checked. There are many typos and grammatical errors. Just a few examples:

line 65. Rhodamine B were used as the target pollutant, should be "was used"

line 143. "a type â…£ cure"   should be “curve”

line 144.  "the pore structure characteristic ...were conducted" should be "characterization ... was conducted"

line 176 “ the adsorption sites on the surface of the sample will increase with higher specific surface area and larger pore volume, which enhanced…”

Future and past tense should be avoided in the same sentence.

Line 182 “we can inferred”

Answer: Thanks for the reviewer’s good advice. We have checked the English carefully. Please check the revised version.

Reviewer 3 Report

The modified manuscript is in better shape. This reviewer recommends accepting this manuscript.

Author Response

Reviewer3#

The modified manuscript is in better shape. This reviewer recommends accepting this manuscript.

Answer: Thanks for the reviewer.

This manuscript is a resubmission of an earlier submission. The following is a list of the peer review reports and author responses from that submission.

Round 1

Reviewer 1 Report

The work has been reported so many times with different research groups. The measured photocatalytic degradation has also been reported already. I do not think the work has any scientific novelty. Nothing new will attract the researchers.

Reviewer 2 Report

The manuscript “molecules-1473852” entitled “Facile synthesis of porous g-C3N4 with enhanced visible-light photo activity” presents interesting synthesis of GCN using mixture of dicyanamide and urea in different combinations. The resultant experiments have shown an increase in surface area of GCN and also resulted in enhanced photo activity. The synthesized Di/Ur catalyst is used for the photo degradation of Rhodamine dye. The enhanced photodegradation activity has been correlated with the better separation of photo generated charge carriers and restarted charge recombination. The proposed GCN catalyst has been found applicable without any metal support and can be used for the degradation of organic pollutants. I recommend this work to be published in Molecules. However few issues/suggestions can be inserted in the revised version to enhance the quality of proposed work. I have a few suggestions below.

  1. Authors can also try one example of colorless organic pollutant in addition to Rhodamine (i.e. colored pollutant).
  2. Photoluminiscence (PL) intensity is related to recombination efficiency of the photocatalyst. Low PL intensity leads to lesser recombination of charge carriers which can be seen in the Figure 7. Can author explain the reason of reduced wavelength in case of Di/Ur catalyst?
  3. Which of the species between superoxide and hydroxyl radical is superior in this metal free catalytic degradation? What about the superior activity in case of CN alone.
  4. In the title visible-light photoactivity is reported but in the section 2.3. Regarding Photoactivity measurements reports use of 600 W Xenon lamp which is a source of UV light. Please clarify?
  5. In Figure 3, authors are suggested to report the binding energy values of CN as well.